# Criteria-Aware Graph Filtering:
# Extremely Fast Yet Accurate Multi-Criteria Recommendation

## ABSTRACT

Multi-criteria (MC) recommender systems, which utilize MC rating information for recommendation, are increasingly widespread in various e-commerce domains. However, the MC recommendation using training-based collaborative filtering, requiring consideration of multiple ratings compared to single-criterion counterparts, often poses practical challenges in achieving state-of-the-art performance along with scalable model training. To solve this problem, we propose CA-GF, a *training-free* MC recommendation method, which is built upon *criteria-aware* graph filtering for *efficient yet accurate* MC recommendations. Specifically, first, we construct an item–item similarity graph using an MC user-expansion graph. Next, we design CA-GF composed of the following key components, including 1) *criterion-specific* graph filtering where the optimal filter for each criterion is found using various types of polynomial low-pass filters and 2) *criteria preference-infused aggregation* where the smoothed signals from each criterion are aggregated. We demonstrate that CA-GF is **(a) efficient**: providing the computational efficiency, offering the extremely fast runtime of less than *0.2 seconds* even on the largest benchmark dataset, **(b) accurate**: outperforming benchmark MC recommendation methods, achieving substantial accuracy gains up to 24% compared to the best competitor, and **(c) interpretable**: providing interpretations for the contribution of each criterion to the model prediction based on visualizations.

## KEYWORDS

Collaborative filtering; criteria preference; graph filtering; low-pass filter; multi-criteria recommender system.

**ACM Reference Format:**
Anonymous Author(s). 2024. Criteria-Aware Graph Filtering: Extremely Fast Yet Accurate Multi-Criteria Recommendation. In *Proceedings of the ACM Web Conference 25 (WWW '25), April 28 – May 02, 2025, Sydney, Australia. August 25-29, 2024, Barcelona, Spain.* ACM, New York, NY, USA, 12 pages. https://doi.org/XXXXXXX.XXXXXXX

## 1 INTRODUCTION

Multi-criteria (MC) recommender systems, which leverage detailed *criteria* ratings for each item, have become increasingly important across various online service areas, including travel, restaurants, hotels, movies, and music [10, 15, 18, 20, 28, 35]. MC recommender systems generally excel in recommending relevant items to users

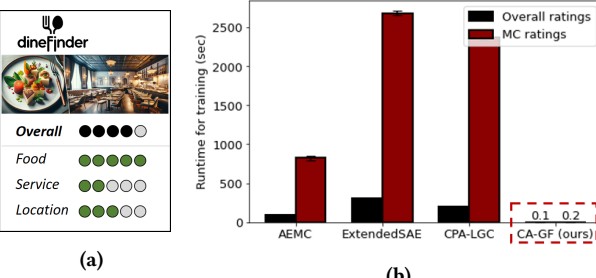

**Figure 1: An illustration showing (a) four criteria ratings in a restaurant domain and (b) a comparison of the training time for 100 epochs between using single-criterion ratings (*i.e.,* overall ratings) and MC ratings across three benchmark MC recommendation methods on the TripAdvisor (TA) dataset. Additionally, the processing time is measured for CA-GF that does not need any training process.**

compared to single-criterion recommender systems, as they capture user preferences more exquisitely [5, 10, 15, 20, 28, 44]. For example, as illustrated in Figure 1a, in a restaurant domain, a user can provide four criteria ratings, which include an *overall* rating as well as other MC ratings for *food*, *service*, and *location*.

Nonetheless, the inclusion of MC ratings in MC recommender systems often significantly increases computational demands compared to single-criterion counterparts, due to the complexity of processing and analyzing multi-dimensional user feedback [16, 39]. For instance, given $N$ user–item interactions with 8 criteria, training-based collaborative filtering (CF) methods having a computational complexity of $O(N^2)$ increase the training time by approximately 64 times over the single-criterion case. As shown in Figure 1b, the time required for identical epochs in training substantially escalates when MC ratings are incorporated into three benchmark deep neural network (DNN)-based MC recommendation methods, including AEMC [28], ExtendedSAE [32], and CPA-LGC [22]. Acknowledging that user preferences evolve quickly due to trends, personal circumstances, and exposure to new content, such latency in training time may not be desirable [1, 12, 23, 25].

Unlike earlier studies on learning models such as graph convolution for recommendations in a parametric manner [7, 8, 37, 38, 43], our study is inspired by recent advances in non-parametric, *i.e.*, training-free, CF for single-criterion recommender systems. More precisely, we leverage the concept of graph signal processing, namely *graph filtering* [17, 29, 40], for CF. While graph filtering offers an affordable solution to the computational challenges in single-criterion recommender systems, it remains open how such methodology can be seamlessly and effectively applied to MC recommender systems without causing computational demands associated with MC ratings. In our study, we aim to develop an innovative

graph filtering-based MC recommendation method, which is efficient yet accurate. To this end, we start by outlining three design **goals (G1–G3)** to attain a successful MC recommender system.

- **G1. Capability of capturing inter-criteria relations:** Canonical MC recommendation methods [4, 5, 20, 28, 44], which simply extend rather popular single-criterion recommendation methods to MC circumstances, often fail to fully grasp the collaborative signal in complex semantics across MC ratings (*i.e.*, inter-criteria relations) [22]. In this context, when MC recommender systems are built upon graph filtering, a natural challenge lies in how to capture *inter-criteria* relations.

- **G2. Awareness of criterion-specific characteristics:** While inter-criteria relation information is exploited in designing MC recommender systems, it is also crucial to preserve the unique characteristics inherent to each criterion. This viewpoint motivates us to comprehensively grasp the *distinctiveness of each criterion*, thus leading to accurate MC recommendations.

- **G3. Fast runtime:** Conventional DNN-based MC recommendation methods are not without the increased training cost caused by making use of MC ratings, compared to the single-criterion counterparts (see Figure 1b). While graph filtering-based CF methods are known to circumvent the computational burden to some extent, it is yet technically challenging how to maximize computational efficiency in devising a graph filtering-based method when MC ratings are concerned.

To effectively achieve **G1**-**G3** at once, we propose CA-GF, a *training-free* MC recommendation method, which accommodates *criteria-aware* graph filters for both efficient and accurate MC recommendations. Specifically, in CA-GF, we initially construct an item–item similarity graph using an MC user-expansion graph to capture complex contextual semantics across the MC ratings (**G1**). Then, based on the item–item similarity graph, we perform *criterion-specific* graph filtering where the optimal filter for each criterion is found using various types of polynomial low-pass filters (LPFs) (**G2**), which do not necessitate costly matrix decomposition in conventional graph filtering-based CF methods [17, 29, 40] (**G3**). Finally, we perform *criteria preference-infused aggregation* to judiciously aggregate the smoothed signals from each criterion (**G2**).

Our main contributions are three-fold and summarized as follows:

(1) **Simply understandable yet effective methodology:** We address the challenge of increased training time caused by MC ratings. To this end, we device CA-GF built upon *polynomial graph filters*. By harnessing criteria awareness using different polynomial LPFs for different criteria, CA-GF is capable of making full use of modern computer hardware (i.e., CPU and GPU), achieving the extremely fast runtime of **less than 0.2 seconds** even on the largest benchmark dataset (*i.e.*, BeerAdvocate (BA)).

(2) **Superior performance:** Through the sophisticated integration of each intra-criterion information as well as inter-criteria relation information into the graph filtering process, CA-GF attains state-of-the-art performance. Our approach consistently outperforms the best competitor by up to **24% in terms of the NDCG@5**.

(3) **Comprehensive empirical studies:** Extensive experiments validate the efficacy of CA-GF, demonstrating its computational efficiency and accuracy. Moreover, unlike traditional DNN-based recommender systems, CA-GF offers clear insights into the model's predictions, providing substantial interpretability benefits.

For reproducibility, the source code of this study is available at https://anonymous.4open.science/r/CA-GF-0D56.

## 2 PRELIMINARIES

### 2.1 Problem Definition

We formally define the top-$K$ MC recommendation, along with basic notations. Let $u \in \mathcal{U}$ and $i \in \mathcal{I}$ denote a user and an item, respectively, where $\mathcal{U}$ and $\mathcal{I}$ denote the sets of all users and all items, respectively. $\mathcal{N}_u \subset \mathcal{I}$ denotes a set of items interacted by user $u$. Compared to single-criterion recommender systems, MC recommender systems comprise of a number of rating criteria. We denote $R_c \in \mathbb{R}^{|\mathcal{U}| \times |\mathcal{I}|}$ as the rating matrix (*i.e.*, the user–item interaction matrix) for criterion $c$. In particular, $R_0 \in \mathbb{R}^{|\mathcal{U}| \times |\mathcal{I}|}$ refers to the overall rating matrix. Then, the top-$K$ MC recommendation problem is formally defined as follows:

*Definition 1.* (Top-$K$ MC recommendation) [22]: Given $u \in \mathcal{U}$ and $i \in \mathcal{I}$, and $C + 1$ user–item rating matrices $R_0 \times R_1 \times \ldots \times R_C$ including an overall rating matrix $R_0$, the top-$K$ MC recommendation aims to recommend top-$K$ items that user $u \in \mathcal{U}$ is most likely to prefer among his/her non-interacted items in $\mathcal{I} \setminus \mathcal{N}_u$ *w.r.t.* the *overall* rating by using all $C + 1$ user–item MC ratings.

### 2.2 Graph Signal Processing

We introduce basic concepts of graph signal processing. First, we consider a weighted undirected graph $G = (V, E)$, represented by an adjacency matrix $A \in \mathbb{R}^{|V| \times |V|}$. A graph signal is a function $f : V \to \mathbb{R}^d$ that encodes the set of nodes into a $d$-dimensional vector $\mathbf{x} = [x_1, x_2, \ldots, x_{|V|}]^T$, where $x_i$ represents the signal strength of node $i$. The smoothness of $\mathbf{x}$ on $G$ is quantified by the graph quadratic form, a measure based on the graph Laplacian $L = D - A$,[1] where $D = \text{diag}(A\mathbf{1})$ is the degree matrix.[2] The smoothness measure $S(\mathbf{x})$ is formally expressed as follows [29, 30]:

$$S(\mathbf{x}) = \sum_{i,j} A_{i,j}(x_i - x_j)^2 = \mathbf{x}^T L \mathbf{x}. \tag{1}$$

The smaller the value of $S(x)$, the smoother the signal $\mathbf{x}$ is on the graph. Next, we formally define the graph Fourier transform (GFT) for a graph signal $\mathbf{x}$ as:

$$\hat{\mathbf{x}} = U^T \mathbf{x}, \tag{2}$$

where $U \in \mathbb{R}^{|V| \times |V|}$ is the GFT basis whose $i$-th column is the eigenvector $u_i$ of $L$ in the eigen-decomposition of $L = U \Lambda U^T$ for $\Lambda = \text{diag}(\lambda_1, \ldots, \lambda_V)$ and ordered eigenvalues $\lambda_1 \leq \cdots \leq \lambda_{|V|}$. Here, the signal $\mathbf{x}$ is considered smooth if the dot product of the eigenvectors corresponding to smaller eigenvalues of $L$ is high. As the GFT is a linear orthogonal transform, the inverse GFT is given

---

[1]Here, we note that the graph Laplacian $L$ can also be defined as its normalized version $L = I - \tilde{A}$, where $\tilde{A} = D^{-1/2} A D^{-1/2}$.

[2]We denote the all-ones vector of any dimension as $\mathbf{1}$ for notational simplicity.

by $\mathbf{x} = U\hat{\mathbf{x}}$. Finally, the graph filter and the graph convolution are formally defined as follows:

**Definition 2.** (Graph filter) [21, 29, 30, 40]: Given a graph Laplacian matrix $L$, a graph filter $H(L) \in \mathbb{R}^{|V| \times |V|}$ is defined as

$$H(L) = U\text{diag}(h(\lambda_1), \ldots, h(\lambda_{|V|}))U^T, \qquad (3)$$

where $h : \mathbb{C} \to \mathbb{R}$ is the frequency response function that maps eigenvalues $\{\lambda_1, \cdots, \lambda_{|V|}\}$ of $L$ to $\{h(\lambda_1), \cdots, h(\lambda_{|V|})\}$.

**Definition 3.** (Graph convolution) [29, 30, 40]: The convolution of a graph signal $\mathbf{x}$ and a graph filter $H(L)$ is given by

$$H(L)\mathbf{x} = U\text{diag}(h(\lambda_1), \ldots, h(\lambda_{|V|}))U^T\mathbf{x}, \qquad (4)$$

which first transforms $\mathbf{x}$ by the GFT and then transforms its filtered signal with $H(L)$ by the inverse GFT.

In signal processing, signals are typically characterized by their smoothness and low-frequency components, whereas noise is usually non-smooth and dominates at high frequencies [29]. In this context, a significant category of filters is LPFs, which enhance the smoothness of graph signals, thereby aiding noise reduction. We refer to Appendix A for the formal definition of LPFs.

## 2.3 Graph Filtering for Single-Criterion Recommendation

LPFs play a crucial role in CF by promoting signal smoothness and reducing high-frequency noise [17, 29]. This is because CF relies on users' ratings, which inherently exhibit low-frequency patterns [17]. Such patterns represent consistent preferences or trends among groups of users. By filtering out noise and smoothing graph signals, LPFs enhance the clarity of the underlying low-frequency patterns, thus leading to more accurate and reliable CF-aided recommendations [17].

Conventional graph filtering-based CF methods [17, 23, 29, 40] in single-criterion recommender systems first construct an item–item similarity graph as in the following:

$$\tilde{P} = \tilde{R}_0^T \tilde{R}_0, \qquad (5)$$

where

$$\tilde{R}_0 = D_U^{-1/2} R_0 D_I^{-1/2}. \qquad (6)$$

Here, $R_0 \in \mathbb{R}^{|\mathcal{U}| \times |\mathcal{I}|}$ is the rating matrix; $\tilde{R}_0$ is the normalized rating matrix; $D_U = \text{diag}(R_0 \mathbf{1})$ and $D_I = \text{diag}(\mathbf{1}^T R_0)$; and $\tilde{P}$ is the adjacency matrix of the item–item similarity graph.

Graph filtering-based CF methods [17, 29, 40] typically employ both linear and ideal LPFs. Representative work includes GF-CF [29], whose graph convolution is formulated as follows:

$$\mathbf{s}_u = \mathbf{r}_u(\tilde{P} + \alpha D_U^{-1/2} \bar{U}\bar{U}^T D_I^{-1/2}), \qquad (7)$$

where $\mathbf{s}_u \in \mathbb{R}^{|\mathcal{I}|}$ is the predicted preferences for user $u$; $\mathbf{r}_u \in \mathbb{R}^{|\mathcal{I}|}$ is the ratings of $u$, which serves as graph signals to be smoothed; $\bar{U} \in \mathbb{R}^{|\mathcal{I}| \times k}$ is the top-$k$ singular vectors of $\tilde{R}_0$; $\tilde{P}$ is the linear LPF in Eq. (5); $D_U^{-1/2} \bar{U}\bar{U}^T D_I^{-1/2}$ is the ideal LPF of $\tilde{P}$; and $\alpha$ is a hyperparameter balancing between the two filters.

A primary benefit of such graph filtering-based CF methods is their non-parametric nature. These approaches bypass the need for intricate and time-intensive model training, relying instead on efficient matrix operations to derive a closed-form solution

corresponding to recommendation scores (*i.e.*, predicted preferences of users, denoted as $\mathbf{s}_u$).

## 3 PROPOSED METHOD: CA-GF

### 3.1 Overview

The objective of our study is to judiciously incorporate MC ratings into graph filtering without losing its computational efficiency. To this end, we propose CA-GF, a not only *training-free* but also *matrix decomposition-free* graph filtering method. In particular, in CA-GF, *criteria-aware* graph filters built upon an MC user-expansion graph are accommodated to effectively capture the collaborative signal in complex contextual semantics across MC ratings.

To achieve the aforementioned goals **G1–G3**, the proposed CA-GF consists of the following components:

(1) **Graph construction:** To accomplish **G1**, CA-GF initially construct an MC user-expansion graph that enables us to capture complex contextual semantics in MC ratings. Next, we construct an item–item similarity graph to design a new graph filtering method with regulated edge weights (see Section 3.2).

(2) **Graph filtering harnessing criteria awareness:** To accomplish **G2**, due to the fact that the optimal filter can be found differently for each criterion, we propose *criterion-specific* graph filtering (see Section 3.3.1). Moreover, we perform *criteria preference-infused aggregation* that combines the smoothed signals from each criterion to enrich the criteria awareness (see Section 3.3.3).

(3) **Polynomial graph filtering:** To accomplish **G3**, we propose to use *polynomial graph filtering* [23], which is performed without costly matrix decomposition to achieve extremely fast recommendation when accommodating multiple graph filters (see Section 3.3.2).

We elaborate on the technical details of the proposed CA-GF method in the following subsections, where the schematic overview is illustrated in Figure 2. We refer to Appendix B for the pseudocode of CA-GF.

### 3.2 Graph Construction

*3.2.1 MC user-expansion graph construction.* As stated in Section 3.1, graph filtering-based CF methods in single-criterion recommender systems [3, 17, 29, 40] typically utilize the rating matrix to construct the item–item similarity graph. However, as long as MC ratings are associated, it is not straightforward how to construct an item–item similarity graph. As the primary component of our study, the first step of CA-GF is to create an *MC user-expansion graph*, where each user is expanded to $C + 1$ different *criterion-user nodes* to capture complex semantics inherent in MC ratings. Precisely, the rating matrix $R_{MC} \in \mathbb{R}^{(C+1)|\mathcal{U}| \times |\mathcal{I}|}$ for our MC user-expansion graph is designed by concatenating the $C + 1$ rating matrices as follows:

$$R_{MC}^T = R_0^T || R_1^T || \ldots || R_C^T, \qquad (8)$$

where the operator $||$ denotes the concatenation of matrices, which enables us to explore complex high-order connectivity among criterion-user nodes and item nodes [22]. Then, we normalize $R_{MC}$ according to the degree of nodes in the graph as in [17, 29, 40]:

$$\tilde{R}_{MC} = D_U^{-1/2} R_{MC} D_I^{-1/2}, \qquad (9)$$

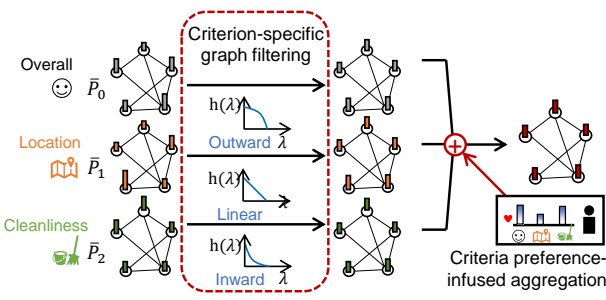

**Figure 2: The schematic overview of CA-GF.**

where $D_U$ and $D_I$ are the diagonal matrices of criterion-user nodes and item nodes, respectively, defined as $D_U = \text{diag}(R_{MC}\mathbf{1})$ and $D_I = \text{diag}(\mathbf{1}^T R_{MC})$.

*3.2.2   Item–item similarity graph with adjustment.* To perform graph filtering, we construct the normalized item-item similarity graph $\tilde{P}_{MC} \in \mathbb{R}^{|I| \times |I|}$ using $\tilde{R}_{MC}$ in Eq. (9) as follows:

$$\tilde{P}_{MC} = \tilde{R}_{MC}^T \tilde{R}_{MC}, \tag{10}$$

which represents the degree of similarity between each pair of items. Note that, compared to the case of expanding each item node to $C+1$ criterion-item nodes [22], our approach effectively prevents the high dimensionality problem of the item–item similarity graph, while achieving **G1** by constructing a unified graph structure across MC ratings towards graph filtering. Next, according to the type of graph filter we use based on $\tilde{P}_{MC}$, the corresponding filtered signals are over-smoothed or under-smoothed, depending on the intensity of connections between nodes in $\tilde{P}_{MC}$. For instance, using a linear LPF (*i.e.*, $\tilde{P}_{MC}$) would be less prone to over-smoothing since it is associated only with the first-order connectivities. Thus, we aim to adjust the filtered signals differently for each graph filter $f(\cdot)$. To this end, similarly as in [23], we employ an additional adjustment process for the graph $\tilde{P}_{MC}$. This process utilizes the Hadamard power $\tilde{P}_{MC}^{\circ s_f}$ by raising each edge weight of $\tilde{P}_{MC}$ to the power of $s_f$, which is formulated as

$$(\bar{P}_f)_{ij} = (\tilde{P}_{MC})_{ij}^{s_f}, \tag{11}$$

where $(\bar{P}_f)_{ij}$ is the $(i, j)$-th element of matrix $\bar{P}_f$ and $s_f$ is the adjustment parameter for graph filter $f(\cdot)$, which will be specified in Section 3.3.

## 3.3   Graph Filtering Harnessing Criteria Awareness

We describe how to perform graph filtering based on the adjusted item–item similarity graph $\bar{P}_f$.

*3.3.1   Criterion-Specific Graph Filtering.* In single-criterion recommender systems, it is required to discover only a single optimal LPF that promotes smoothness of the graph signals for denoising [29]. In MC recommendations, however, the optimal LPF for each criterion is often different. For instance, for relatively subjective criteria (*spec.*, *price* in a hotel domain), low-frequency components should be utilized less than those in other criteria. In our study, to

attain **G2**, we propose *criterion-specific* graph filtering, which applies *diverse LPFs* for different criteria to smooth out graph signals as shown in Figure 2. The predicted rating $s_{u,c} \in \mathbb{R}^{|I|}$ of user $u$ for criterion $c$ is then characterized as:

$$\mathbf{s}_{u,c} = \mathbf{r}_{u,c} f(\bar{P}_f, c), \tag{12}$$

where $\mathbf{r}_{u,c}$ is the $u$-th row of $R_c$, which will be used as graph signals of user $u$ for criterion $c$; and $f(\bar{P}_f, c)$ is the graph filter *specific to criterion $c$* for graph $\bar{P}_f$. Here, Eq. (12) is the signal $\mathbf{r}_{u,c}$ convolved with $f(\bar{P}_f, c)$ (refer to Definition 3).

*3.3.2   Polynomial Graph Filtering.* We now specify the criterion-specific graph filter $f(\bar{P}_f, c)$ in Eq. (12) that is decided depending on each criterion $c$. As stated in **G3**, using multiple filters for MC ratings naturally produces additional computation costs caused by a matrix decomposition process in Eq. (7). To bypass the high computation overhead, a recent study [23] proposed to use *polynomial graph filters* for CF. Inspired by this, we also employ multiple polynomial graph filters, applying a distinct polynomial LPF for each criterion. The polynomial graph filter up to the $K$-th order can be expressed as

$$f(\bar{P}_f, c) = \sum_{k=1}^{K} a_{c,k} \bar{P}_f^k, \tag{13}$$

where $f(\bar{P}_f, c)$ is the polynomial graph filter specific to criterion $c$; $a_{c,k}$ is the coefficient of a matrix polynomial; and $K$ is the maximum order of the matrix polynomial basis. Note that, using Eq.(13), any LPFs can be designed by adjusting $a_{c,k}$. The following lemma states that the design of universal LPFs is established.

LEMMA 4. *[23, 29]: The matrix polynomial $\sum_{k=1}^{K} a_{c,k} \bar{P}_f^k$ is a graph filter for graph $\bar{P}_f$, with the frequency response function of $h(\lambda) = \sum_{k=1}^{K} a_{c,k}(1 - \lambda)^k$.*

According to Lemma 4, one can find the optimal polynomial LPF by extensively searching for $\{a_{c,k}\}_{k=1}^{K}$ using the validation set, which however comes at the expensive computation costs and thus violates our design goal **G3**. Alternatively, we present three representative (*i.e.*, predefined) polynomial LPFs, namely linear ($\bar{P}_f$), inward ($\bar{P}_f^2$), and outward ($2\bar{P}_f - \bar{P}_f^2$) LPFs, which essentially embrace a broad set of LPFs. These three types of graph filters are implemented within second-order polynomials (*i.e.*, $K = 2$). As depicted in Figure 2, the three polynomial LPFs behave differently, affecting how much low-frequency components are exploited compared to the rest of the spectrum. We note that one can use other types of LPFs with matrix polynomials based on one's own design choice. Using Lemma 4, we theoretically show how each polynomial LP graph filter has its unique frequency response function as follows. All proofs are provided in Appendix C.

COROLLARY 4.1.   *(**Linear LPF**) [17, 29]: The matrix $\bar{P}_f$ is equivalent to a linear LPF with the following frequency response function*

$$h(\lambda) = 1 - \lambda. \tag{14}$$

COROLLARY 4.2.   *(**Inward LPF**) The matrix $\bar{P}_f^2$ is equivalent to an inward LPF of the item-item similarity graph $\bar{P}_f$ with the following frequency response function*

$$h(\lambda) = \lambda^2 - 2\lambda + 1. \tag{15}$$

COROLLARY 4.3. **(Outward LPF)** *The matrix* $2\bar{P}_f - \bar{P}_f^2$ *is equivalent to an outward LPF of the item-item similarity graph* $\bar{P}_f$ *with the following frequency response function*

$$h(\lambda) = 1 - \lambda^2. \tag{16}$$

From Corollaries 4.1–4.3, we pay attention to choosing optimal $f(\bar{P}_f, c)$ out of the three polynomial LPFs for each criterion $c$. Here, it is worth noting that our polynomial LPFs can be implemented through simple matrix calculations, which thereby allows us to more effectively leverage well-optimized machine learning and computation frameworks such as PyTorch [24] and CUDA [27] to enhance computational speed via parallel computation. As shown in Figure 1b, such operation makes CA-GF immune to the computational burden as the number MC ratings increases, thereby resulting in successfully achieving **G3**. Moreover, thanks to such rapid computation of polynomial graph filtering, the optimal filter for each criterion is easily found using the validation set. For the detailed analysis of computational complexity of CA-GF, we refer to Appendix D.

*3.3.3 Criteria Preference-Infused Aggregation.* One can aggregate the smoothed signals in Eq. (12) by simply summing up the smoothed signals for all criteria. However, users often exhibit different criteria preferences [22, 31]. For instance, in a hotel domain, a user tends to make decisions based on the cleanliness aspect of a hotel, while another user decides based on the service aspect. To accommodate such personalized information into our CA-GF method, we additionally devise a novel *criteria preference-infused aggregation* technique for elaborately capturing the criteria preferences of each user. From the fact that the number of ratings often differs from each criterion [22], we claim that, if a user gave more and/or higher ratings on a certain criterion, then he/she tends to reveal a higher preference for the criterion, which will be empirically validated in Section 4.4. Based on this claim, we formalize our aggregation technique as follows:

$$\hat{C} = \tilde{X}\bar{T}; \bar{T}_{ij} = T_{ij}^{s_f}; T = \tilde{X}^T\tilde{X};$$
$$\tilde{X} = XD_X^{-1}X = (R_0\mathbf{1})||(R_1\mathbf{1})|| \cdots ||(R_C\mathbf{1}); \tag{17}$$

where $X \in \mathbb{R}^{|U|\times(C+1)}$ represents the sum-rating matrix for each criterion whose $(u, c)$-th element refers to the sum of ratings given by a user $u$ to all relevant items for criterion $c$; $\tilde{X}$ is the normalized matrix of $X$ along $D_X^{-1} = \text{diag}(X\mathbf{1})$; $\bar{T} = \tilde{X}^T\tilde{X} \in \mathbb{R}^{(C+1)\times(C+1)}$ is the criterion–criterion similarity graph; $\bar{T}_{ij}$ is the $(i, j)$-th element of the adjacency matrix of criterion–criterion similarity graph $\bar{T}$ adjusted with the parameter $s_f$; and $\hat{C} \in \mathbb{R}^{|U|\times(C+1)}$ is the criteria preference matrix in which $\tilde{X}$ serves as signals for graph filtering. Then, the matrix $\hat{C}$ is used for weights during aggregation to infuse the criteria preferences of users. Finally, as illustrated in Figure 2, the predicted rating of user $u$ after the criteria preference-infused aggregation is expressed as

$$\mathbf{s}_u = \frac{1}{C+1}\sum_{c=0}^{C}\hat{C}_{u,c}\mathbf{s}_{u,c} = \frac{1}{C+1}\sum_{c=0}^{C}\hat{C}_{u,c}\mathbf{r}_{u,c}f(\bar{P}_f, c), \tag{18}$$

where $\hat{C}_{u,c}$ is the preference of user $u$ on the criterion $c$ in $\hat{C}$.

**Table 1: Statistics of the three datasets used in our experiments.**

|  | # of users | # of items | # of overall ratings | # of MC ratings | $C$ | Density |
|---|---|---|---|---|---|---|
| TA | 5,132 | 7,205 | 41,638 | 280,521 | 7 | 0.11% |
| YM | 1,827 | 1,471 | 46,239 | 231,195 | 4 | 1.72% |
| BA | 10,726 | 10,832 | 626,995 | 3,134,981 | 4 | 0.54% |

## 4 EXPERIMENTAL EVALUATION

In this section, we systematically conduct extensive experiments to address the key research questions (RQs) outlined below:

- **RQ1 (Efficiency):** How fast is CA-GF compared to benchmark MC recommendation methods in terms of runtime?
- **RQ2 (Accuracy):** How much does CA-GF improve top-$K$ recommendation accuracy over benchmark recommendation methods?
- **RQ3 (Ablation study):** How does each component in CA-GF contribute to the recommendation accuracy?
- **RQ4 (Sensitivity):** How do key parameters affect the performance of CA-GF?
- **RQ5 (Interpretability):** How precisely does CA-GF provide interpretations relevent to the criteria?

### 4.1 Experimental Settings

**Datasets.** We carry out experiments on three public datasets, which are widely used in studies on MC recommendation [5, 16, 20, 22, 28, 32]: TripAdvisor (TA), Yahoo!Movie (YM), and BeerAdvocate (BA). Statistics of the three datasets are summarized in Table 1.

**Competitors.** To comprehensively demonstrate the superiority of CA-GF, we present six benchmark MC recommendation methods (including ExtandedSAE [32], AEMC [28], DMCF [20], CFM [4, 5], CPA-LGC [22], and GF-CF$_{MC}$). Additionally, to observe the potential benefits of MC ratings, we include four representative single-criterion recommendation methods (namely NGCF [37], LightGCN [7], GF-CF [29], and DiffRec [36]), where only overall ratings are used due to the incapability of using MC ratings as input. In our study, to show the results by a naïve extension of GF-CF to MC settings, we additionally introduce a variant of GF-CF, termed GF-CF$_{MC}$. This variant employs GF-CF [29] to each of $C + 1$ item–item similarity graphs constructed from MC ratings, and then aggregates the output through summation for the final prediction. We refer to Appendix E.2 for further details of GF-CF$_{MC}$.

**Evaluation protocols.** We randomly select 80% of the interactions of each user for the training set and the remaining 20% as the test set. From the training set, we randomly select 10% of interactions as the validation set for hyperparameter tuning. To evaluate the accuracy of top-$K$ MC recommendation, we use benchmark metrics that are widely used in literature [6–8, 33, 37, 38, 42, 43], such as *recall* and *normalized discounted cumulative gain (NDCG)*, where $K$ is set to 5 and 10 by default. In the test phase, we treat user–item interactions with *overall ratings* that are higher than the median rating in the test set as positive, following the protocols in other studies on the MC recommendation [20, 22, 28, 32]. For each metric, we report the average taken over 10 independent runs except for deterministic methods (*i.e.*, GF-CF, GF-CF$_{MC}$, and CA-GF).

**Implementation details.** We use the best hyperparameters of competitors obtained by extensive hyperparameter tuning on the

**Table 2: Runtime (in seconds) and recommendation accuracy (in NDCG@10) on the largest dataset (BA).**

| Method | Training | NDCG@10 | Runtime (s) |
|---|---|---|---|
| ExtendedSAE | ✓ | 0.0052 | 11,520 |
| AEMC | ✓ | 0.0752 | 1,022 |
| CPA-LGC | ✓ | 0.1396 | 10,020 |
| GF-CF$_{MC}$ | ✗ | 0.1344 | 274 |
| CA-GF | ✗ | **0.1477** | **0.2** |

validation set. Due to space limitation, we specify default hyperparameters of CA-GF in Appendix E.3. Unless otherwise stated, for all training-based methods, we set the dimensionality of the embedding to 64 and use the Adam optimizer [13], where the batch size is set to 128. All experiments are carried out with the same device for a fair comparison: Intel (R) 12-Core (TM) i7-9700K CPUs @ 3.60 GHz and GPU of NVIDIA GeForce RTX A6000.

### 4.2 RQ1: Efficiency Analysis

Table 2 showcases the computational efficiency on the BA dataset, the largest dataset with more than $3M$ MC ratings.[3] Moreover, to validate the scalability of CA-GF, we present a runtime comparison on different device configurations, i.e., cases without GPU (i.e., CPU only) and with GPU, using various synthetic datasets with $C = 4$ in Figure 3;[4] in this experiment, we generated seven synthetic datasets whose sparsity level is controlled to 98.5%, where the numbers of (users, items, MC ratings) are set to $\{(1.5K, 3K, 0.3M), (2.5K, 5.5K, 1M), (4K, 6K, 1.8M), (5K, 9K, 3.4M), (8K, 10K, 6M), (10K, 15K, 11M), (25K, 20K, 38M)\}$. Our findings are as follows:

(i) Notably, Table 2 demonstrates that training-free methods, GF-CF$_{MC}$ and CA-GF, outperform training-based methods such as ExtendedSAE, AEMC, and CPA-LGC in terms of runtime efficiency. Specifically, CA-GF achieves the *runtime of 0.2 seconds*, whereas other training-based methods require over $1,000$ seconds for the model convergence.

(ii) Furthermore, Table 2 shows that, although GF-CF$_{MC}$ alleviates the computational demands of matrix decomposition using the generalized power method [11, 29], CA-GF is over $2,160\times$ faster than GF-CF$_{MC}$ on the BA dataset. This is owing to the use of matrix decomposition-free polynomial filters in CA-GF, which efficiently utilize GPU resources for achieving **G3**. Moreover, Figure 3 demonstrates that, while GF-CF$_{MC}$ is a training-free solution, CA-GF consistently and significantly outperforms GF-CF$_{MC}$ in terms of runtime across the datasets of various sizes.

(iii) Figure 3 reveals the scalability of CA-GF. By fully utilizing the parallel computing capabilities of GPU, CA-GF runs consistently within 2 seconds for datasets containing up to $6M$ MC ratings. Even when the size of the loaded data exceeds GPU memory limits, CA-GF demonstrates robust performance on CPU, maintaining runtime below 2 minutes for datasets exceeding $38M$ MC ratings. This highlights the computational efficiency of CA-GF, especially in handling large-scale datasets.

---

[3]We refer to Appendix E.4 for the empirical analysis of efficiency on other datasets.
[4]GF-CF$_{MC}$ was not implemented on GPU as utilizing GPU for calculating ideal LPFs is not straightforward.

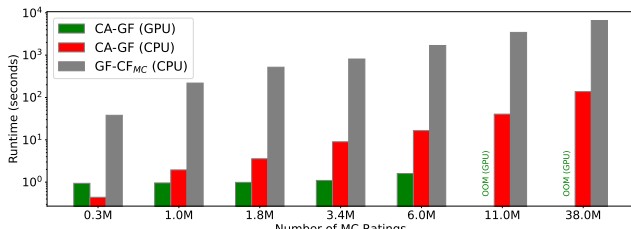

**Figure 3: Log-scaled runtime comparison of CA-GF (with and without GPU) and GF-CF$_{MC}$ (CPU) using various synthetic datasets, where OOM (GPU) indicates out-of-memory issues on GPU.**

**Table 3: Performance comparison among CA-GF and all competitors for the three benchmark datasets. Here, MC represents the methods using MC ratings. The best and second-best performers are highlighted in bold and underline, respectively.**

| Method | MC | Metric | K | TA | YM | BA |
|---|---|---|---|---|---|---|
| NGCF | - | Recall@K | 5 | 0.0370 | 0.0855 | 0.0551 |
| | | | 10 | 0.0513 | 0.1352 | 0.0925 |
| | | NDCG@K | 5 | 0.0310 | 0.1056 | 0.1255 |
| | | | 10 | 0.0363 | 0.1183 | 0.1275 |
| LightGCN | - | Recall@K | 5 | 0.0550 | 0.0970 | 0.0566 |
| | | | 10 | 0.0724 | 0.1502 | 0.0937 |
| | | NDCG@K | 5 | 0.0512 | 0.1154 | 0.1320 |
| | | | 10 | 0.0586 | 0.1325 | 0.1298 |
| GF-CF | - | Recall@K | 5 | 0.0596 | 0.1217 | 0.0645 |
| | | | 10 | 0.0728 | 0.1753 | 0.1034 |
| | | NDCG@K | 5 | 0.0570 | 0.1440 | 0.1333 |
| | | | 10 | 0.0612 | 0.1574 | 0.1344 |
| DiffRec | - | Recall@K | 5 | 0.0574 | 0.0834 | 0.0695 |
| | | | 10 | 0.0637 | 0.1003 | 0.1030 |
| | | NDCG@K | 5 | 0.0527 | 0.1123 | 0.1230 |
| | | | 10 | 0.0637 | 0.1420 | 0.1392 |
| ExtendedSAE | ✓ | Recall@K | 5 | 0.0024 | 0.0766 | 0.0018 |
| | | | 10 | 0.0048 | 0.1150 | 0.0044 |
| | | NDCG@K | 5 | 0.0012 | 0.0912 | 0.0026 |
| | | | 10 | 0.0025 | 0.1001 | 0.0052 |
| AEMC | ✓ | Recall@K | 5 | 0.0538 | 0.0802 | 0.0353 |
| | | | 10 | 0.0664 | 0.1112 | 0.0588 |
| | | NDCG@K | 5 | 0.0530 | 0.0909 | 0.0737 |
| | | | 10 | 0.0574 | 0.0990 | 0.0752 |
| DMCF | ✓ | Recall@K | 5 | 0.0312 | 0.0333 | 0.0307 |
| | | | 10 | 0.0388 | 0.0470 | 0.0411 |
| | | NDCG@K | 5 | 0.0334 | 0.0541 | 0.0312 |
| | | | 10 | 0.0401 | 0.0614 | 0.0401 |
| CFM | ✓ | Recall@K | 5 | 0.0411 | 0.0420 | 0.0375 |
| | | | 10 | 0.0501 | 0.0613 | 0.0531 |
| | | NDCG@K | 5 | 0.0357 | 0.0392 | 0.0891 |
| | | | 10 | 0.0398 | 0.0538 | 0.0920 |
| CPA-LGC | ✓ | Recall@K | 5 | 0.0630 | 0.1211 | 0.0625 |
| | | | 10 | 0.0830 | 0.1725 | 0.0966 |
| | | NDCG@K | 5 | 0.0550 | 0.1392 | 0.1388 |
| | | | 10 | 0.0650 | 0.1532 | 0.1396 |
| GF-CF$_{MC}$ | ✓ | Recall@K | 5 | 0.0617 | 0.1223 | 0.0643 |
| | | | 10 | 0.0753 | 0.1755 | 0.1038 |
| | | NDCG@K | 5 | 0.0595 | 0.1457 | 0.1329 |
| | | | 10 | 0.0645 | 0.1588 | 0.1344 |
| CA-GF | ✓ | Recall@K | 5 | **0.0750** | **0.1224** | **0.0704** |
| | | | 10 | **0.0854** | **0.1765** | **0.1144** |
| | | NDCG@K | 5 | **0.0738** | **0.1476** | **0.1464** |
| | | | 10 | **0.0774** | **0.1608** | **0.1477** |

### 4.3 RQ2: Recommendation Accuracy

We compare the accuracy of CA-GF against competitors specified in Section 4.1. For the methods that were originally designed for

**Table 4: The performance comparison among CA-GF and its four variants in terms of the Recall@10. Here, the best performer is highlighted in bold.**

|          | TA         | YM         | BA         |
| -------- | ---------- | ---------- | ---------- |
| CA-GF    | **0.0854** | **0.1765** | **0.1147** |
| CA-GF-m  | 0.0718     | 0.1751     | 0.1031     |
| CA-GF-s  | 0.0713     | 0.1698     | 0.1076     |
| CA-GF-f  | 0.0725     | 0.1751     | 0.1077     |
| CA-GF-p  | 0.0840     | 0.1760     | 0.1143     |

single-criterion recommender systems (NGCF, LightGCN, GF-CF, and DiffRec), we use single ratings (*i.e.*, overall ratings). Table 3 shows the results of all the competitors and CA-GF. Our findings are as follows:

(i) CA-GF consistently outperforms all the competitors regardless of the performance metrics and datasets. In particular, CA-GF achieves state-of-the-art performance with significant gains up to 24% in terms of the NDCG@5 on TA.

(ii) The use of MC ratings remarkably boosts the performance, with CPA-LGC and CA-GF surpassing their single-criterion counterparts, namely LightGCN and GF-CF, respectively.

(iii) MC recommendation methods that are built upon DNNs or matrix factorization (ExtendedSAE, AEMC, DMCF, and CFM) show comparatively inferior performance to that of GCN (CPA-LGC) or graph filtering (GF-CF$_{MC}$ and CA-GF). This implies that explicitly exploiting the complex structural information for MC recommendations is indeed beneficial for accurate recommendations.

(iv) Meanwhile, performance comparison between CA-GF and GF-CF$_{MC}$ reveals that, while graph filtering-based methods generally yield superior results, the gain of CA-GF over GF-CF$_{MC}$ is also significant. This underscores that the mere adoption of graph filtering for MC recommendations is insufficient to achieve optimal accuracy.

## 4.4 RQ3: Ablation Studies

To analyze the contribution of each component in CA-GF, we conduct an ablation study in comparison with four variants depending on which components are taken into account for designing the end-to-end CA-GF method. The performance comparison among the four methods is presented in Table 4 *w.r.t.* the Recall@10 using three datasets.

- CA-GF: corresponds to the original CA-GF method without removing any components;
- CA-GF-m: removes the MC user-expansion graph. That is, only a single criterion (*i.e.*, overall ratings) is used for graph construction;
- CA-GF-s: replaces all $s_f$'s by 1. That is, the adjustment parameter is ablated;
- CA-GF-f: sets all $f(\bar{P}_f, c)$'s to the linear LPF in Eq. (14). That is, only a single polynomial LPF is used;
- CA-GF-p: removes criteria preference-infused aggregation. Instead, all the smoothed signals from each criterion are evenly aggregated by simple summation.

Our observations are as follows:

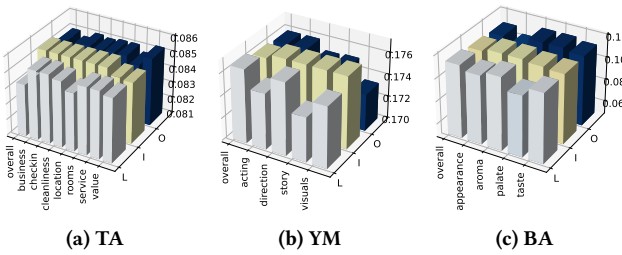

(a) TA      (b) YM      (c) BA

**Figure 4: The effect of three polynomial LPFs $f(\bar{P}_f, c)$ (L, I, and O) for each criterion on the Recall@10.**

(i) All four modules in CA-GF plays a crucial role in the success of the proposed CA-GF method.

(ii) The performance gap between CA-GF and CA-GF-f reveals that using diverse polynomial LPFs is important for accurate MC recommendation.

(iii) The performance gap between CA-GF and CA-GF-s tends to be much higher than that between CA-GF and other variants except for the BA dataset. This finding indicates that the use of proper adjustment parameters for graph filter $f(\bar{P}_f, c)$ is crucial for accurate recommendations.

(iv) Albeit slightly, the gain of CA-GF over CA-GF-p manifests a positive impact of criteria preference-infused aggregation on the recommendation accuracy for all the datasets.

## 4.5 RQ4: Sensitivity Analysis

We investigate the impact of key parameters in CA-GF on the recommendation accuracy, which include the selection of graph filters $f(\bar{P}_f, c)$ for criterion $c$ and the adjustment parameter $s_f$ for each $f(\bar{P}_f, c)$. For notational simplicity, we denote the three polynomial LPFs (*i.e.*, linear, inward, and outward LPFs) as L, I, and O, respectively. When each parameter varies so that its effect is revealed, other parameters are set to the pivot values specified in Appendix E.3.

**(Effect of $f(\bar{P}_f, c)$)** From Figure 4, it is seen that using different filters for each criterion $c$ produces different recommendation accuracies, which confirms that each criterion needs its specific optimal graph filter (see Section 3.3.1). Here, we again note that the optimal filter for each criterion is found quite promptly, thanks to the minimal computation time required for CA-GF.

**(Effect of $s_f$)** Figure 5 shows how the Recall@10 behaves according to different values of $s_f$ for each of three polynomial LPFs (L,I, and O). Our findings are as follows:

(i) The performance is not uniformly sensitive across different $s_f$'s. Some regimes of $s_f$ exhibit high sensitivity, whereas others show low sensitivity.

(ii) For L, the optimal $s_f$ is found in $s_f < 1$ except for YM, which is the densest dataset. For I and O, the opposite pattern is observed while the optimum lies mostly in the range of $s_f > 1$. This is because I and O, implemented using the second-order polynomial of $\tilde{P}$, are more prone to either over-smoothing, where predicted ratings become overly similar by exploring the collaborative signal in higher-order connectivities. On the other hand, since L employs only the first-order polynomial of $\tilde{P}$, it is less prone to over-smoothing.

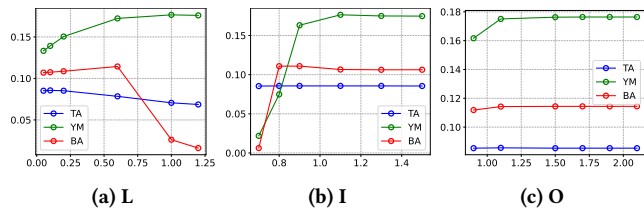

(a) L                                    (b) I                                    (c) O

**Figure 5: The effect of adjustment parameter $s_f$ for three polynomial LPFs (L, I, and O) on the Recall@10.**

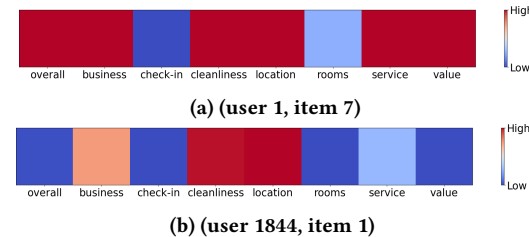

(a) (user 1, item 7)

(b) (user 1844, item 1)

**Figure 6: Attribution maps that visualize the contribution of each criterion to CA-GF's predictions for the TA dataset.**

(iii) This variation is partly due to the use of different orders of $\tilde{P}$ for each polynomial LPF. Hence, in performance optimization, it is of utmost importance to select an appropriate value of $s_f$ for each $f(\bar{P}_f, c)$. For example, for I and O, it is desirable to use $s_f > 1$ to effectively mitigate the over-smoothing problem by enhancing value distinctions.

### 4.6 RQ5: Interpretability

Unlike canonical black-box recommendation methods based on DNNs, CA-GF offers significant advantages in model interpretability, allowing an in-depth understanding of each prediction process. For instance, we generate an attribution map that quantifies the influence of MC on the model predictions, Specifically, the attribution map of the user–item interaction $(u, i)$ visualizes the $i$-th component of $\hat{C}_{u,c}s_{u,c} \in \mathbb{R}^{|\mathcal{I}|}$ in Eq. (18) for different $c$'s. It represents each user's criteria preferences for certain items, which provides insight into the importance of each criterion. For instance, Figure 6 illustrates two attribution maps for the two different interactions of (user, item), including (1, 7) and (1844, 1) on the TA dataset. The following observations are made.

(i) These two instances display different patterns, which verifies that each criterion makes a different contribution to the model prediction.

(ii) It is likely that the two criteria *check-in* and *rooms* contribute less to the prediction of CA-GF, compared to the other criteria.

This level of interpretability is invaluable, not just for comprehending the model's functionality but also for guiding strategic business decisions, thus enabling model refinement and enhancement. We refer to Appendix E.4 for further visualization results on other datasets.

## 5 RELATED WORK

**Graph filtering-based approaches.** Within the realm of graph signal processing, GCN is regarded as a parameterized graph convolutional filter [14, 29]. As a representative approach, NGCF [37] was introduced by learning appropriate LPFs while capturing high-order collaborative signals inherent in user–item interactions [29]. LightGCN [7] achieved convincing performance by eliminating linear transformation and non-linear activation from the GCN layers in NGCF. By closing the gap between LightGCN and graph filtering methods alongside a closed-form solution for the infinite-dimensional LightGCN, GF-CF [29] stood out for achieving accurate recommendation performance while significantly reducing time consumption with its parameter-free nature. FIRE [40] was introduced by improving this approach to an incremental recommendation method. Additionally, PGSP [17] made use of a mixed-frequency filter that combines a linear LPF with an ideal LPF.

**MC recommender systems.** Efforts have consistently been made to incorporate MC ratings to enhance the accuracy of recommendations. In one of the initial endeavors, a support vector regression-based approach [9] was introduced to assess the relative importance of individual criteria ratings. CFM [5] was formulated by collectively employing matrix factorization for MC rating matrices. DTTD [2] was developed by integrating cross-domain knowledge alongside side information. Moreover, in light of the extensive adoption of deep learning, there has been a continuous endeavor to develop DNN-based MC recommender systems. For instance, ExtendedSAE [32] was introduced to capture the relationship between MC ratings using stacked autoencoders. LatentMC [15] was designed with variational autoencoders to map user reviews into latent vectors, constituting latent MC ratings. DMCF [20] was devised for predicting MC ratings using a DNN, with the predicted ratings being aggregated by another DNN. AEMC [28] employed deep autoencoders, capturing non-linear relationships between users' preferences for criteria. As pioneer work on integrating light graph convolution into MC recommender systems, CPA-LGC [22] was devised to capture complex semantics in MC ratings.

## 6 CONCLUSIONS AND FUTURE WORK

In this paper, we explored how fast and accurate graph filtering can be developed in MC recommender systems. To this end, we introduced CA-GF, the first attempt to design a graph filtering-based MC recommendation method that is not only training-free but also decomposition-free, thereby circumventing the problem of the computational burden that MC ratings entail. Through extensive experiments on three benchmark datasets, we demonstrated the impact and benefits of CF-GF from various perspectives, including (a) the extraordinarily computational efficiency with the runtime of less than 1 second on BA, the largest dataset, (b) the superior accuracy over other competing MC recommendation methods, (c) the impact of using different optimal LPFs for each criterion, (d) the effectiveness of each component, and (e) the interpretability via visualizing each user's criteria preferences for certain items. Potential avenues of our future research include the design of an adaptive graph filter such that the optimal LPF for each criterion is found more effectively.

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

## A DEFINITION OF LPF

We formally define the LPF as follows:

*Definition 5.* (LPF) [26, 29, 34]: For $k = 1, \cdots, |V|$ and $\lambda_1 \leq \cdots \leq \lambda_{|V|}$, the graph filter $H(L)$ is $k$-low-pass if and only if $\eta_k \in [0, 1]$, where

$$\eta_k := \frac{max\{|h(\lambda_{k+1})|, \cdots, |h(\lambda_{|V|})|\}}{min\{|h(\lambda_1)|, \cdots, |h(\lambda_k)|\}}. \tag{19}$$

## B PSEUDOCODE OF CA-GF

We summarize the end-to-end process of CA-GF in Algorithm 1.

---

**Algorithm 1** CA-GF

---

**Input:** MC ratings $R_0 \times R_1 \times ... \times R_C$, set of users $\mathcal{U}$, set of items $\mathcal{I}$
**Output:** $\mathbf{s}_u$ for $u \in \mathcal{U}$
1: $R_{MC}^T = R_0^T || R_1^T || \ldots || R_C^T$
2: $\tilde{R}_{MC} = D_U^{-1/2} R_{MC} D_I^{-1/2}$
3: $\tilde{P} = \tilde{R}_{MC}^T \tilde{R}_{MC}$
4: Calculate $\bar{P}_f$ for each $f(\cdot)$ using Eq. (11)
5: Calculate $\hat{C}$ using Eq. (17)
6: **for** $u \leftarrow 0$ to $|\mathcal{U}|$ **do**
7:    **for** $c \leftarrow 0$ to $C$ **do**
8:       $\mathbf{s}_{u,c} = \mathbf{r}_{u,c} f(\bar{P}_f, c)$
9:    **end for**
10:    $\mathbf{s}_u = \frac{1}{C+1} \sum_{c=0}^{C} \hat{C}_{u,c} \mathbf{s}_{u,c}$
11: **end for**
12: **return** $\mathbf{s}_u$ for $u \in \mathcal{U}$

---

## C PROOFS OF COROLLARIES

**Proof of Corollary 4.1.** Although the proof of Corollary 4.1 is presented in [17], we provide the full proof for completeness. First, the Laplacian matrix $L$ can be decomposed as

$$L = U\Lambda U^T = U\text{diag}(\lambda_1, \lambda_2, \ldots, \lambda_m)U^T, \tag{20}$$

which indicates that $L$ is a graph filter with the frequency response function of $h(\lambda) = \lambda$. Then, we have

$$L = I - \bar{P}_f. \tag{21}$$

Suppose that $U_i$ is the eigenvector of $L$ corresponding to the eigenvalue $\lambda_i$, Then, using Eq. (21), it follows that

$$LU_i = \lambda_i U_i = (I - \bar{P}_f)U_i, \tag{22}$$

resulting in

$$\bar{P}_f U_i = (1 - \lambda_i)U_i. \tag{23}$$

This means that $L$ and $\bar{P}_f$ have the same eigenvectors and the corresponding eigenvalues have the following relationship:

$$(\lambda_{\bar{P}_f})_i = 1 - \lambda_i, \tag{24}$$

where $(\lambda_{\bar{P}_f})_i$ is the $i$-th largest eigenvalue of $\bar{P}_f$. Suppose

$$\Lambda_{\bar{P}_f} = \text{diag}((\lambda_{\bar{P}_f})_1, (\lambda_{\bar{P}_f})_2, \ldots, (\lambda_{\bar{P}_f})_m). \tag{25}$$

Then, we have

$$\bar{P}_f = U\Lambda_{\bar{P}_f}U^T = U\text{diag}(1 - \lambda_1, 1 - \lambda_2, \ldots, 1 - \lambda_n)U^T, \tag{26}$$

which means that $\bar{P}_f$ is the polynomial LPF with the frequency response function of

$$h(\lambda) = 1 - \lambda, \tag{27}$$

which completes the proof of the Corollary 4.1.

**Proof of Corollary 4.2.** The symmetric matrix $\bar{P}_f$ can be decomposed as $\bar{P}_f = U\Lambda_{\bar{P}_f}U^T$, where $\Lambda_{\bar{P}_f}$ is a diagonal matrix of the eigenvalues of $\bar{P}_f$. Now, due to the fact that $U^T = U^{-1}$, $\bar{P}_f^2$ becomes

$$\bar{P}_f^2 = (U\Lambda_{\bar{P}_f}U^T)(U\Lambda_{\bar{P}_f}U^T) = U\Lambda_{\bar{P}_f}^2 U^T. \tag{28}$$

Using the relation $\Lambda_{\bar{P}_f} = I - \Lambda$ from Corollary 4.1, we have

$$\bar{P}_f^2 = U(I - \Lambda)^2 U^T. \tag{29}$$

Using the expansion

$$(I - \Lambda)^2 = I - 2\Lambda + \Lambda^2, \tag{30}$$

Eq. (29) can be rewritten as

$$\bar{P}_f^2 = U(I - 2\Lambda + \Lambda^2)U^T, \tag{31}$$

which indicates that $\bar{P}_f^2$ plays a role of mapping each eigenvalue $\lambda_i$ of $\bar{P}_f$ to $\lambda_i^2 - 2\lambda_i + 1$. Hence, the frequency response function of $\bar{P}_f^2$ is given by

$$h(\lambda) = \lambda^2 - 2\lambda + 1, \tag{32}$$

which completes the proof of Corollary 4.2.

**Proof of Corollary 4.3.** Given $\bar{P}_f = U\Lambda_{\bar{P}_f}U^T$, we have

$$\begin{aligned}2\bar{P}_f - \bar{P}_f^2 &= 2U\Lambda_{\bar{P}_f}U^T - U\Lambda_{\bar{P}_f}^2 U^T \\ &= U(2\Lambda_{\bar{P}_f} - \Lambda_{\bar{P}_f}^2)U^T.\end{aligned} \tag{33}$$

Using the relation $\Lambda_{\bar{P}_f} = I - \Lambda$ from Collorary 4.1, we have

$$2\Lambda_{\bar{P}_f} - \Lambda_{\bar{P}_f}^2 = 2(I - \Lambda) - (I - \Lambda)^2 = I - \Lambda^2. \tag{34}$$

Thus, Eq. (33) can be rewritten as

$$2\bar{P}_f - \bar{P}_f^2 = U(I - \Lambda^2)U^T, \tag{35}$$

which indicates that $2\bar{P}_f - \bar{P}_f^2$ contributes to mapping each eigenvalue $\lambda_i$ of $L$ to $1 - \lambda_i^2$. Hence, $2\bar{P}_f - \bar{P}_f^2$ is the polynomial LPF of $\bar{P}_f$ with the frequency response function of

$$h(\lambda) = 1 - \lambda^2, \tag{36}$$

which completes the proof of Corollary 4.3.

**Table 5: Runtime on the three benchmark datasets.**

|  | ExtendedSAE | AEMC | CPA-LGC | GF-CF$_{MC}$ | CA-GF |
|---|---|---|---|---|---|
| TA | 2,657 | 871 | 2,345 | 316 | **0.2** |
| YM | 45 | 27 | 731 | 34 | **0.03** |
| BA | 11,520 | 1,022 | 10,020 | 274 | **0.2** |
| Training | ✓ | ✓ | ✓ | ✗ | ✗ |

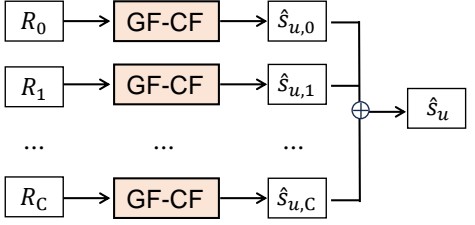

**Figure 7: The schematic overview of GF-CF$_{MC}$.**

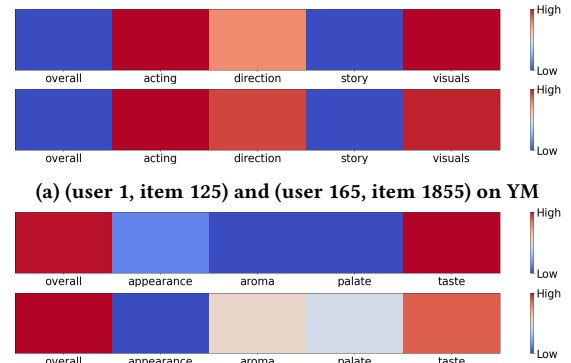

**(a) (user 1, item 125) and (user 165, item 1855) on YM**

**(b) (user 10, item 1544) and (user 4895, item 1382) on BA**

**Figure 8: Attribution maps that visualize the contribution of each criterion to CA-GF's predictions for the (a) YM and (b) BA datasets.**

## D COMPLEXITY ANALYSIS

We provide a theoretical analysis of CA-GF's computational complexity. Although the complexities of sparse matrix multiplication can vary, we base our analysis on the complexity of $O(\text{nnz})$ [41], where nnz represents the non-zero components. CA-GF comprises two primary steps: graph construction and filtering. First, for the graph construction step $P = R_{MC}^T R_{MC}$, the nnz of $R_{MC}$ is equal to the number of MC ratings (, denoted as $n_{mc}$, hence $O(n_{mc})$. Next, in the graph filtering stage $R_{MC}P$, the nnz of $R_{MC}$ is $n_{mc}$, and while the nnz of $P = R_{MC}^T R_{MC}$ is not explicitly expressible to $n_{mc}$, it is equivalent to the total number of user pairs sharing common items, denoted as $n_P$. Typically, $n_{mc} < n_P$ and $n_P$ are not significantly large in scale compared to $n_{mc}$; therefore, we assert that the final computational complexity for graph filtering is $O(n_P)$.

In consequence, the computational complexity of CA-GF is **linear** in nnz in the polynomial graph filter used, which is generally not large due to the high sparsity characteristic of recommendation datasets. Additionally, using parallel GPU computation for matrix multiplication, CA-GF achieves constant runtime regardless of the dataset size, provided sufficient GPU VRAM is available.

## E DETAILS OF EXPERIMENTS

### E.1 Dataset Description

We describe the details of the datasets used in our experiments.
**TripAdvisor (TA)**: The TA dataset, released by [35], comprises hotel rating information, including an overall rating as well as ratings for seven comprehensive criteria: *business, check-in quality, cleanliness, location, rooms, service,* and *value*. The ratings are on a scale of 1 to 5 for all criteria.
**Yahoo!Movie (YM)**: The YM dataset, first introduced by [10], comprises movie rating information, including an overall rating as well as ratings for four specific criteria: *story, acting, direction,* and *visuals*. The ratings are on a scale of 1 to 5 for all criteria.
**BeerAdvocate (BA)**: The BA dataset, released by [18, 19], comprises beer rating information, including an overall rating as well as ratings for four specific criteria: *appearance, aroma, taste,* and *palate*. The ratings range from 1 to 5 for all criteria.

### E.2 Details of GF-CF$_{MC}$

Figure 7 illustrates the schematic overview of GF-CF$_{MC}$. Given MC rating matrices $R_c$ for criterion $c \in \{0, 1, \cdots, C\}$, GF-CF$_{MC}$ first separately construct $C + 1$ bipartite graphs, and performs GF-CF [29] on each of $C + 1$ different criteria. Then, the predicted ratings $\hat{s}_{u,c}$ for each criterion are aggregated by simple summation for the final prediction. This approach differs from CA-GF, which uses an integrated graph, called the MC user-expansion graph, for graph filtering. Moreover, while GF-CF$_{MC}$ uses the same graph filter across the criteria, CA-GF leverages a different optimal graph filter depending on the criterion, which effectively harnesses criteria awareness.

### E.3 Default Hyperparameters of CA-GF

We describe default hyperparameters of CA-GF, each of which is tuned on the validation set. The hyperparameters of CA-GF are listed as follows: 1) the polynomial LPF $f(\bar{P}_f, c)$; 2) the adjustment parameter $s_f$ used for the item–item similarity graph $\bar{P}_{MC}$; and 3) the adjustment parameter $s$ used for the criterion–criterion similarity graph $\tilde{T}$. First, the polynomial LPF $f(\bar{P}_f, c)$ optimally found for all criteria on each dataset is as follows:

- TA: {*overall*: O, *business*: L, *check-in*: I, *cleanliness*: L, *location*: L, *rooms*: I, *service*: L, *value*: L};
- YM: {*overall*: O, *acting*: I, *direction*: I, *story*: O, *visuals*: L };
- BA: {*overall*: L, *appearance*: O, *aroma*: I, *palate*: I, *taste*: L}.

Second, the adjustment parameter $s_f$ for each graph filter is set as

- TA: {L: 0.1, I: 1, O: 1.2};
- YM: {L: 1, I: 1, O: 1.8};
- BA: {L: 0.6, I: 0.85, O: 1.5}.

Lastly, the adjustment parameter $s_f$ for the criterion–criterion similarity graph is set as {TA: 2, YM: 4, BA: 2}. For the graph construction of each dataset in GNN-based competitors (NGCF, LightGCN, and CPA-LGC), edges are included if their ratings are higher than the median value.

## E.4 Experimental Results on Other Datasets

First, Table 5 showcases the (average) runtime on all the datasets including BA. Here, the runtime of training-based methods (ExtendedSAE, AEMC, and CPA-LGC) refers to the total time spent on model training for 100 epochs, whose duration is empirically determined to be sufficient for model convergence. Second, in Figure 8, we show the experimental results corresponding to RQ5 on the YM and BA datasets, along with the attribution maps visualizing the contribution of each criterion to the model prediction.

