# OpenReview forum: "Criteria-Aware Graph Filtering: Extremely Fast Yet Accurate Multi-Criteria Recommendation"
_ACM.org/TheWebConf/2025/Conference — WWW 2025 Poster_

### Official Review · Reviewer_HsHt · 2024-11-28

**Novelty:** 5
**Technical Quality:** 5

**Review:**

The paper presents a novel multi-criteria (MC) recommendation system, CA-GF, which leverages criteria-aware graph filtering to provide efficient and accurate recommendations. The authors have addressed a significant gap in the field by proposing a training-free and matrix decomposition-free approach, which is particularly innovative given the computational challenges associated with MC ratings.

**Questions:**

1. The paper could provide more insights into the methodology for determining the choices of hyper-parameter, particularly for the polynomial LPFs.
2. A deeper comparative analysis with single-criterion recommendation methods and the variant CA-GF-m may be helpful to the further understanding of CA-GF.
3. Inconsistent results exist in Tables 3 and 4.

**Reviewer Confidence:**

2: The reviewer is willing to defend the evaluation, but it is likely that the reviewer did not understand parts of the paper

**Scope:**

3: The work is somewhat relevant to the Web and to the track, and is of narrow interest to a sub-community

---

### Official Review · Reviewer_xSFa · 2024-12-01

**Novelty:** 6
**Technical Quality:** 6

**Review:**

This paper addresses the problem of multi-criteria recommendation and introduces a training-free graph filtering algorithm for inferring item rankings while considering multiple criteria. The proposed method is highly efficient, as it eliminates the need for a training process, and achieves superior performance compared to GNN-based baselines and matrix decomposition methods across three datasets.

The proposed method is interesting and offers valuable insights into graph filtering-based collaborative filtering. Intuitively, learning on user-item bipartite graphs without additional node or edge features does not necessitate too complex modeling and can potentially capture user preferences through connectivity signals.

I do not find significant issues with the methodology or experiments; however, I have a few questions and concerns:

1. In the Introduction (Lines 93-94), I wonder why a training-based algorithm has a computational complexity of O(N^2) where N is the number of criteria. A straightforward approach is to aggregate and average the representations across the criteria graphs, which would result in O(N) complexity since it conducts the aggregation N times. Please also provide more details on how GNN-based baselines were adapted to the multi-criteria setup.
2. The intuition behind certain design choices is not clearly explained. For instance, why does raising the power of edge weights (Equ.(11) and Equ.(17)) help alleviate the over-smoothing issue? The process of obtaining $\hat{C}$ is not entirely straightforward and would benefit from further elaboration.
3. The performance appears sensitive to hyperparameter selection. It would be helpful to provide a criterion or guidelines for hyperparameter tuning to improve reproducibility.
4. The codebase presented in the paper is not available.

**Questions:**

See Review section.

**Reviewer Confidence:**

2: The reviewer is willing to defend the evaluation, but it is likely that the reviewer did not understand parts of the paper

**Scope:**

3: The work is somewhat relevant to the Web and to the track, and is of narrow interest to a sub-community

---

### Official Review · Reviewer_NH5m · 2024-12-02

**Novelty:** 4
**Technical Quality:** 4

**Review:**

This paper proposes a novel training-free graph filtering-based method (CA-GF) for multi-criteria (MC) recommendation tasks. CA-GF achieves impressive computational efficiency and competitive accuracy by utilizing polynomial low-pass filters (LPFs) and an MC user-expansion graph. The authors have presented experimental results in the outperformance and efficiency, which look promising.

Pros:
1)	Clear motivation and easy to follow.
2)	Experiments support the claims in the introduction.
3)	The computational efficiency in MC recommendation is remarkably outstanding.
Cons:
1)	Fails to compare with recent state-of-the-art methods, including those from 2024, which could better highlight the novelty and competitiveness of the proposed approach.
2)	The Related Work section primarily enumerates existing works but lacks a comprehensive analysis of their differences and relevance to the proposed method.
3)	The Figure 2 lacks clarity and requires a detailed explanation of each component.
4)	Many of the references cited in the paper are relatively old, and the format needs to be standardized, such as [6], [11], [20] lacking page numbers.

**Questions:**

1)	The paper employs three types of polynomial low-pass filters (linear, inward, and outward), but it does not provide an analysis of why these filters are suitable for multi-criteria recommendation. Is polynomial graph filtering applicable to all types of multi-criteria recommendation problems?
2)	The paper evaluates the method on BA dataset with a large number of ratings. However, does the method yield similar performance when applied to datasets with a large number of users?

**Reviewer Confidence:**

4: The reviewer is certain that the evaluation is correct and very familiar with the relevant literature

**Scope:**

4: The work is relevant to the Web and to the track, and is of broad interest to the community

---

### Official Review · Reviewer_X1uk · 2024-12-03

**Novelty:** 3
**Technical Quality:** 3

**Review:**

This paper proposes a multi-criteria (MC) recommendation method called CA-GF that leverages criteria-aware graph filtering for efficient yet accurate MC recommendations. The key components of CA-GF include criterion-specific graph filtering using optimal polynomial filters for each criterion, and criteria preference-infused aggregation to combine the smoothed signals from each criterion. CA-GF is shown to be computationally efficient, providing extremely fast runtime of less than 0.2 seconds even on large datasets, while also outperforming benchmark MC recommendation methods by up to 24% in accuracy. Additionally, CA-GF offers interpretability by providing insights into the contribution of each criterion to the model predictions. The research topic is quite interesting and the strategy designed is efficient. The paper is well-organized and presented well.

The key contributions are:
1. Proposes a training-free and decomposition-free MC recommendation method called CA-GF that uses criteria-aware graph filtering
2. Better computational efficiency and accuracy over other competing MC recommendation methods.
3. Interpretability via visualizing each user’s criteria preferences for certain items

Yet it can still be improved, as:
1. Insufficient academic contribution: The CA-GF algorithm looks like a combination of existing schemes, e.g., the Graph construction part of the algorithm is borrowed from [1][2], and the graph filtering part also looks like a direct application of polynomial graph filtering on MC recommendation.
2. The authors' assumptions in the Criteria Preference-Infused Aggregation section are rather arbitrary and the slight enhancement in the results of the ablation experiments is not enough to validate the assumptions. Perhaps further theoretical proof and argumentation might make the design of the section more convincing.
3. The ablation experiment on MC user-expansion graph can incorporate another design similar to GF-CF_MC to verify the validity of the MC user-expansion graph construction design. The current experiment can only verify the effect of “Multi-Criteria” on the results.

To summarize, this paper has an interesting research topic, and the proposed algorithm is effective and efficient, but the contribution is limited, and some of the designs of the algorithm lack convincing validation.

[1] Park, J. D., Li, S., Cao, X., & Shin, W. Y. (2023, August). Criteria tell you more than ratings: Criteria preference-aware light graph convolution for effective multi-criteria recommendation. In Proceedings of the 29th ACM SIGKDD Conference on Knowledge Discovery and Data Mining (pp. 1808-1819).

[2] Park, J. D., Shin, Y. M., & Shin, W. Y. (2024, July). Turbo-cf: Matrix decomposition-free graph filtering for fast recommendation. In Proceedings of the 47th International ACM SIGIR Conference on Research and Development in Information Retrieval (pp. 2672-2676).

**Questions:**

Please refer to the review part.
I have updated new comments for this paper.

**Reviewer Confidence:**

3: The reviewer is confident but not certain that the evaluation is correct

**Scope:**

4: The work is relevant to the Web and to the track, and is of broad interest to the community